# Free Methylglyoxal as a Metabolic New Biomarker of Tumor Cell Proliferation in Cancers

**DOI:** 10.3390/cancers16233922

**Published:** 2024-11-22

**Authors:** Dominique Belpomme, Stéphanie Lacomme, Clément Poletti, Laurent Bonesso, Charlotte Hinault-Boyer, Sylvie Barbier, Philippe Irigaray

**Affiliations:** 1Department of Cancer Clinical Research, Paris V University Hospital, 75005 Paris, France; 2European Cancer and Environment Research Institute (ECERI), 1000 Brussels, Belgium; philippei.artac@gmail.com; 3Centre de Ressources Biologiques, BB-0033-00035, CHRU, 54500 Nancy, France; s.lacomme@chru-nancy.fr; 4Laboratoire Bioavenir, 57000 Metz, France; clem.poletti@gmail.com (C.P.); sbarbierkittler@icloud.com (S.B.); 5Clinical Chemistry Laboratory, Pasteur University Hospital, 06000 Nice, France; bonesso.l@chu-nice.fr (L.B.); hinault-boyer.c@chu-nice.fr (C.H.-B.); 6Université Côte d’Azur, INSERM U1065, C3M, 06000 Nice, France; 7Association for Research on Treatment Against Cancer (ARTAC), 75015 Paris, France

**Keywords:** methylglyoxal, cancer, biomarker, blood, staging, diagnostic, tumor proliferation

## Abstract

This manuscript describes an innovative biomarker for cancer. Cancer cells often exhibit altered metabolism, known as the Warburg effect, where they preferentially use anaerobic glycolysis even in the presence of oxygen. This leads to increased production of methylglyoxal, a side-product of glycolysis. We demonstrate here for the first time that human tumor cells can produce and release free methylglyoxal at high levels, whereas normal cells do not. Consequently, it appears that, in spite of some limitations, free methylglyoxal can be used as a metabolic new clinically useful biomarker in cancers, including cases for which there is still no available biomarker. Our findings open the way to further bio-clinical developments.

## 1. Introduction

Cancer is the second highest cause of death worldwide, with lung, colon, breast (female), pancreas and prostate (male) cancers being most common [1]. With the growing number of cancer cases diagnosed each year worldwide and the overall persisting high number of deaths [2], the identification of new biomarkers for both early detection and targeted therapeutic interventions is sorely needed, to provide better outcomes for patients.

Since 1998, biomarkers used in clinical practice and research have been defined by the US National Institutes of Health (NIH) Biomarkers Definition Working Group [3], and more recently by the Food and Drug Administration (FDA) –NIH Biomarkers, EndpointS, and other Tools (BEST) Working Group [4], as indicators that can be routinely used and repeatedly measured to objectively characterize diseases and their development. In oncology, such biomarkers could be produced either by the tumor itself or by the body in response to the malignant pressure on normal cells, which become cancerous. However, a major challenge in harnessing the potential role of cancer biomarkers for early detection is that cancer initiation and promotion and tumor progression are complex processes involving various abnormal genetic and epigenetic events and cellular interactions [5,6].

In addition, cancer may result from individual risk factors [7] and from exposure to many and diverse environmental carcinogenic agents, such as chemicals, radiation and/or microorganisms, especially in genetically susceptible hosts [8,9,10]. Consequently, tumors vary widely in their etiology and pathogenesis, so that many bioassays were unsuccessful in evidencing correlation between clinical endpoints and tumor biomarkers; thus, there is still a lack of clinically useful biomarkers for many tumors to aid oncologists in primary prevention, decision-making and patient care.

A fundamental property of cancer cells is their metabolic reprogramming, allowing them to increase glucose uptake and glycolysis [11,12]. Thus, it appears that methylglyoxal (MG), a side-product of glycolysis, is a fundamental metabolic biomarker, whose levels may discriminate between cancer cells and normal cells.

As reported in a previous experimental study using a rat colon adenocarcinoma model [13], we showed that free MG blood levels increase significantly in animals grafted with a PRO tumorigenic cell clone (*p* = 0.003), while animals grafted with a REG non-growing tumor cell clone derived from the same initial chemically-induced tumor remain with normal free MG levels [14]. In addition, this clearly demonstrates that, in the animals grafted with the PRO tumorigenic clone, free MG blood levels correlate positively with tumor growth and proliferation, at least in this rat model.

We now show that free MG measured in the peripheral blood of cancer patients is a metabolic new clinically useful biomarker of tumor cell growth and proliferation that allows cancer diagnosis, prognosis, targeted therapeutic decision-making, and the follow-up of patients. In addition, using in vivo experimental methods, we show that free MG can be recovered from human malignant tumors at significantly higher levels than from cells of their corresponding normal tissues, and that human tumor cells produce and release free MG in the peripheral blood.

## 2. Materials and Methods

In order to confirm our rat experimental data [14] in a clinical setting, we prospectively measured free MG in 139 evaluable patients bearing various types of cancer at different TNM stages and compared them to 68 healthy normal controls. Moreover, we showed that free MG blood levels do not correlate with age in healthy subjects (ρ = 0.111; *p* = 367), nor in cancer patients (ρ = −0.0134; *p* = 0.8929), and do not differ in sex in healthy subject (*p* = 0.502), nor in patients with cancer (*p* = 0.1367). We also compared the results obtained in cancer patients with those of 12 normo-glycemic treated type 2 diabetes cases, and to 10 hyperglycemic untreated or relapsed type 2 diabetes cases (for demographic data, see Table 1).

TNM staging was obtained in 133 cancer patients and therapeutic response in 98 patients. The non-availability of all 139 cancer patients for TNM staging and therapeutic response was due to a lack of clinical information. For therapeutic response assessment, we used internationally recognized standards [15]. Complete responses were obtained after surgery and/or radiotherapy in 30 evaluable cases with limited cancer extension (TNM stages I and II), and in 8 cases with advanced disease (TNM stages III or IV);,while partial response was obtained in 31 cases and stable/progressive response in 29 cases, mainly after chemotherapy and/or radiotherapy. Moreover, using repeated free MG peripheral blood measurements, a follow-up of the 38 patients with complete response was carried out.

To validate biologically our clinical data, we also measured free MG in the tumor and in the corresponding normal lung tissue, and peripheral blood of 18 patients with large cell bronchus carcinoma before any treatment.

Our prospective research study involving humans has been approved by the ethical committee of Île de France II for blood samples (Trial Registration Number: 2007-03-03), and by the ethical committee of Est III for surgical tumor and normal lung tissue samples and simultaneous blood samples in cancer patients (Trial Registration Number: DC-2008-459).

### 2.1. MG Measurement

In this clinical study, blood samples were collected on heparin. For practical reasons, we used the whole blood of patients and normal controls instead of plasma, because early centrifugation at 4 °C by nurses was not possible. All whole blood samples were then frozen to −80 °C, before being treated for free MG quantification.

The method used for the measurement of free MG blood levels is that proposed by Rabbani and Thornalley in 2014 [16], with some modifications described in the Appendix A. For tumors and normal tissues, investigations in patients with large cell bronchus carcinoma samplings were realized during surgery. Biopsies of normal lung tissue were carried out distantly to the tumor. Quantification of free methylglyoxal in tumors and their corresponding normal tissue was performed after disruption and homogenization of tumor and normal tissue.

### 2.2. Statistical Analysis

We used the Wilcoxon rank sum test, as we found that our data were not in accordance with a normal distribution, for comparison between the different MG values obtained in patients and healthy controls. Because multiple comparisons were carried out, we used the Bonferroni statistical correction, which adapts the α cut-off determination. We used also the Pearson product–moment correlation test to measure the strength of the association between free MG levels in healthy subjects and age, free MG levels in patients with bronchus or digestive tumors cancer and age, free MG levels and TNM stages, and free MG levels in tumors and in the blood of patients. All statistical analyses were conducted using XLSTAT 2021.1 software.

## 3. Results

### 3.1. Clinical Data

As depicted in Table 2, compared to healthy subjects or normo-glycemic treated type 2 diabetic patients, mean blood level values of free MG are significantly elevated in cancer patients overall (*p* < 0.0001). By contrast, free MG blood level mean values in cancer patients do not significantly differ from those in hyperglycemic untreated or relapsed type 2 diabetes patients used as positive control (*p* = 0.965).

Figure 1 shows the significant positive correlation between free MG blood levels and the different TNM stage categories of cancer patients (ρ: 0.5639; *p* < 0.0001). Whatever the TNM stage of cancer, except for stage I patients, all free MG blood level values were found to be above the mean normal control value of 0.06 µM, and the highest free MG blood level values were observed in stage IV cancer cases. This shows that systematic measurement of free MG in the blood of non-diabetic cancer patients may be a complementary tool for diagnosis, staging and prognostic assessment.

Table 3 shows the mean free MG blood level values +/− standard errors and confidence interval limits according to the different localizations of cancers, such as large cell bronchus, glioblastoma, breast (female), prostate (male), colorectal, pancreas, gynecologic and head and neck carcinoma. Compared to healthy controls with the exception of head and neck carcinoma, all cancer localizations showed a significant higher mean free MG level (*p* between 0.002 and *p* < 0.0005). This was particularly the case for large cell bronchus carcinoma, pancreas carcinoma and glioblastoma for which there is presently no available blood tumor biomarker.

Figure 2 shows that the mean free MG blood level values are above the mean normal reference value of 0.06 µM in most of the investigated types of cancer. There were, however, several cases under the mean normal MG values in breast (female), prostate (male) and head and neck carcinoma. Accordingly, these cases should be considered as false negative cases.

Furthermore, Table 4 shows that cancer patients with complete therapeutic response were associated with normal free MG blood levels, whereas patients with partial or no response have persisting elevated free MG blood levels, suggesting that, in addition to currently available cancer biomarkers and imaging techniques, measurement of free MG in the blood of cancer patients could contribute to the evaluation of therapeutic response and follow-up of patients. Indeed, during the follow-up of the 38 patients with complete therapeutic response, thanks to repeated free MG measurements, we were able to detect a tumor relapse in 10 patients earlier than with the use of other clinical and/or biological investigation methods.

### 3.2. Experimental Data

To justify the above reported clinical data, we measured free MG levels recovered from the tumors of 18 patients with large cell bronchus carcinoma and compared their intra-tumoral free MG levels with those of their corresponding normal lung tissue. We found that the amount of free MG per mg of protein in tumor reached 58.70 +/− 11.98 nmoles, whereas it was 38.98 +/− 7.40 nmoles in normal lung tissue, a highly significant difference (*p* < 0.0001). This finding therefore shows that in vivo human tumor cells produce a significantly higher amount of free MG than cells from normal tissue.

Furthermore, Figure 3 shows the positive correlation obtained for the 18 patients investigated between the tumor/normal free MG level tissue ratio, and the corresponding free MG level in the peripheral blood of each patient (ρ: 0.8245; *p* < 0.0001). This strongly suggests that free MG produced in tumors is released in the peripheral blood of patients.

## 4. Discussion

In recent years, cancer has emerged as a metabolic disease, where cancer cells adapt and proliferate in different tissue micro-environments. However, changes in cancer cell metabolism cannot be explained solely by their adaptation to the tumor microenvironment. They also depend on genetic reprogramming, i.e., on mutations in tumor-promoting oncogenes and/or in tumor suppressor genes, and/or epigenetic changes driving cancer cells to over-express glucose transporters and glycolytic enzymes [17,18,19].

Because anaerobic glycolysis occurs faster than mitochondrial oxidative-phosphorylation (OXPHOS) [20,21], cancer cells rely on anaerobic glycolysis, even in the presence of normal intra-tumoral oxygen availability. This was first reported in 1920 by Otto Warburg and termed aerobic glycolysis [22]. The Warburg effect, according to which cancer cells rely on increased glucose consumption and lactate production in aerobic conditions, has been confirmed in many tumors, including breast, bronchus and colorectal carcinomas and melanomas [23]. Today a major application of the Warburg effect is the development and clinical use of 18F-fluoro-deoxy-glucose (18F-FDG) Positron Emission Tomography (PET) imaging in these cancers to localize tumor and monitor tumor evolution [24]. Note that, in case of increased free MG detection, 18F-FDG PET could be used for tumor localization.

Considered a key hallmark of cancer [12], aerobic glycolysis has been shown to be associated with the production of the side-product, and highly reactive, dicarbonyl MG. We know that MG is mainly generated through the non-enzymatic degradation of triose phosphate glycolytic intermediates, namely dihydro-acetone phosphate and glyceraldehyde-3-phosphate [25]. Due to its electrophilic properties, MG is a potent glycating agent of proteins, nucleic acids and lipids [26], leading to MG-Advanced Glycation End (AGE) products, such as MG-hydroimidazolones and MG-Argypirimidines, which are commonly formed in cancers [27], and which can decrease the diverse normal molecular cell functions by combining with macro-molecules [28,29].

An important consequence of the glycolytic increase in cancer cells is that a major part of the increased intracellular MG level forms AGEs, while a minor part of it is released in the form of permeable and diffusible free MG molecules in the extracellular compartment [26].

In this study we demonstrate for the first time that human tumor cells can produce and release in vivo higher free MG levels than corresponding tissue normal cells. However, since proliferating normal cells can also produce free MG at high levels [30], it cannot be excluded that the increased free MG levels we previously obtained experimentally [14], and in the present study in the blood of cancer patients may be produced by and released both from cancer and stromal cells, and that the positive correlation we obtained between the increase of free MG blood levels and tumor growth may have been caused by the increased number of both types of proliferative cells.

Indeed, it is tempting to speculate that stromal fibroblasts in a tumor can also produce and release MG, since it has been demonstrated that, through a reverse Warburg effect, stromal fibroblasts, under the influence of cancer cells, can be glycolytic and produce and release both free MG and lactate in order to drive and fuel tumor growth and proliferation [31,32].

Confirming our previous laboratory animal experimental data [14], we have shown that repetitive free MG measurement in the blood of cancer patients can be used clinically to assess tumor growth and proliferation in many types of cancer; making free MG a new clinically useful metabolic biomarker in oncology. Moreover, since free MG peripheral blood level normalization occurred in the case of complete therapeutic response in cancer patients, whereas patients who did not respond completely to treatment were associated with persisting high free MG levels, this biomarker could be used for the therapeutic follow-up of patients. Note that we found no significant difference between patients with partial response and patients with stable/progressive tumors. Therefore, repeated free MG level measurements could be a more precise new tool than the other usual methods for therapeutic monitoring in cancer patients.

A biological basis of our clinical data may come from previous findings that extracellular MG-AGEs can activate the receptor for AGEs (RAGE), which is expressed on many cells, including cancer cells [33,34,35]. Indeed, we must clearly distinguish the intracellular effects of MG-AGEs from the extracellular ones. While the precise intracellular MG-AGE concentrations that delineate a pro- or anti-tumor effect are not yet clear, and lead to the concept of a dual role of MG [36], measuring free MG in the blood reveals the possibility of an increase in circulating MG-AGEs in cancer patients. Using cell cultures such as activation of RAGE by MG-AGEs has been well established in many types of cancers, showing that RAGE activation inhibits apoptosis [37], induces inflammation and neo-angiogenesis via the vascular endothelial growth factor (VEGF) [26], and finally promotes tumor growth and metastasis via pathways such as AP-1, NF-kB, PI3K and mToR [26].

Therefore, the pro-cancer role of circulating MG-AGEs/RAGE activation and its downstream signaling cascades may explain our clinical data, since free MG can make covalent combinations with circulating proteins in the blood.

To counteract the harmful electrophilic effect of MG, mammalian cells have developed several cytosolic detoxifying enzymes, among which glyoxalases I (Glo-1) and glyoxalases II (Glo-2) constitute the most important MG detoxification system. It has been shown that Glo-1 is a rate limiting inducible enzyme that uses reduced glutathione (GSH) as cofactor to catalyze the conversion of hemithioacetal formed by the reaction between MG and GSH into S-D-lactoylglutathione, while Glo-2 hydrolyzes S-D-lactoylglutathione in D-Lactate, thereby regenerating GSH [38].

We know that Glo-1 is over-expressed in many types of cancers [39,40], possibly as a consequence of the higher accumulation of MG in cancer cells [41], while Glo-2 expression might be decreased [42,43,44], meaning that the defect in MG detoxification may cause a saturation of the intracellular accumulation of MG, and consequently a release of free MG in the extracellular compartment.

Glo-1 over-expression in cancer has been considered as an independent risk factor associated with poor prognostic and multi-drug resistance [45]. We have confirmed this in breast carcinoma [46], but it could not be evidenced clinically in colorectal carcinoma [47], in which low Glo-1 activity was found to be associated with poor prognostic high stage tumors [47]. It has been emphasized that this ambivalent role of Glo-1 as a tumor promotor or suppressor might be cancer-type dependent. In fact, measurement of the expression of Glo-1 without Glo-2 in tumors may be insufficient to assess prognosis. Moreover, Glo-1 and Glo-2 measurement in tumors, as well as by any tumor immuno-histochemical analysis, cannot be used repeatedly for the follow-up of patients. This explains why diverse assays have proposed the use of repeated measurements of circulating AGEs or AGE-derived products as cancer biomarkers to establish prognosis in cancers [48,49,50], but, in contrast and to our knowledge, these attempts have not yet demonstrated any usefulness in clinical oncology.

There are, however, several limitations to our results.

First, in Table 1, our control healthy group was associated with a median and mean age lower than that of our cancer group, which confirms that cancer arises later in patients. We, however, have shown that there was no correlation between free MG level and age in normal healthy controls (ρ: = 0.111; *p* = 367), as well as in cancer patients (ρ = −0.0134; *p* = 0.8929).

Second, as indicated in Figure 1 and Figure 2, free MG level values were under the normal mean reference value of 0.06 µM. This was particularly the case for TNM stage I evaluable cases, but not so in more extended TNM stages (Figure 1). There were also several cases under the reference value for the different types of cancer investigated (Figure 2). Consequently such cases constitute false negative cases.

Third, since increased free MG in the blood of patients is not specific to cancer, there may be false positive results. As previously reported [26,51] and confirmed in Table 2, MG is also highly produced in hyperglycemic conditions, with high free MG levels in the blood of patients with type 2 diabetes. Before considering free MG as a cancer biomarker, it is therefore mandatory to exclude any MG contribution from diabetes mellitus.

Likewise, accumulation of MG-AGE has been shown to be a characteristic feature of aging and to contribute to the development of neurodegenerative diseases, such as Alzheimer’s disease [26,52]. However, in our study, we did not find any correlation between free MG and age in cancer patients, as well as in normal healthy controls; yet in hypertension and cardiovascular diseases, AGEs have been shown to accumulate in blood, collagen and kidney [26]. To our knowledge, it is not clear however whether these pathologies could be associated with increased circulating free MG levels above normal values. Nevertheless, it appears clear that increased free MG detected in the blood of any patient could be a false positive diagnosis of cancer.

Finally, we must focus on the technical difficulty in treating the human blood samples since, after drawing, they need to be kept below 4 °C.

## 5. Conclusions

In spite of these limitations, it appears from our study that free MG can be used as a metabolic new biomarker in cancers, including cases for which there is presently no available biomarker. Our findings open the way to further bio-clinical developments in this necessary advancement in medicine.

## Figures and Tables

**Figure 1 cancers-16-03922-f001:**
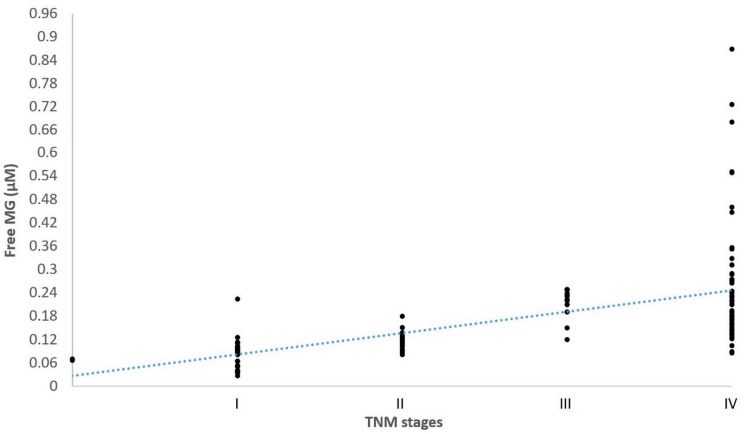
Positive correlation between free MG blood levels and the different TNM stage categories in 133 investigated cancer patients. The blue dotted line represent the linear regression line.

**Figure 2 cancers-16-03922-f002:**
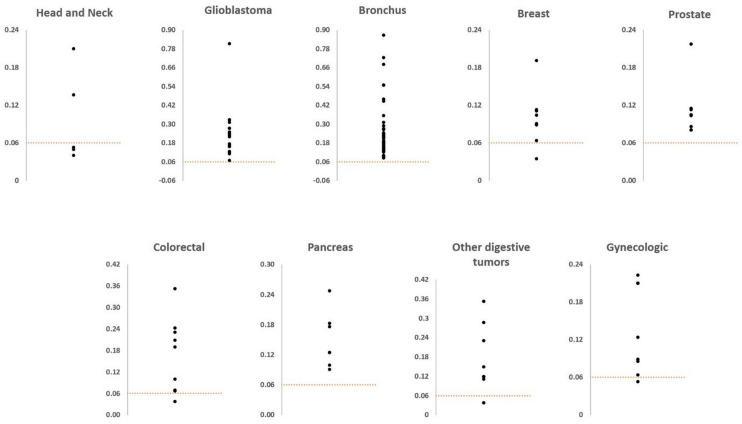
Free MG blood levels in the different categories of cancer investigated. The dotted line refers to the mean normal control value of 0.06 µM.

**Figure 3 cancers-16-03922-f003:**
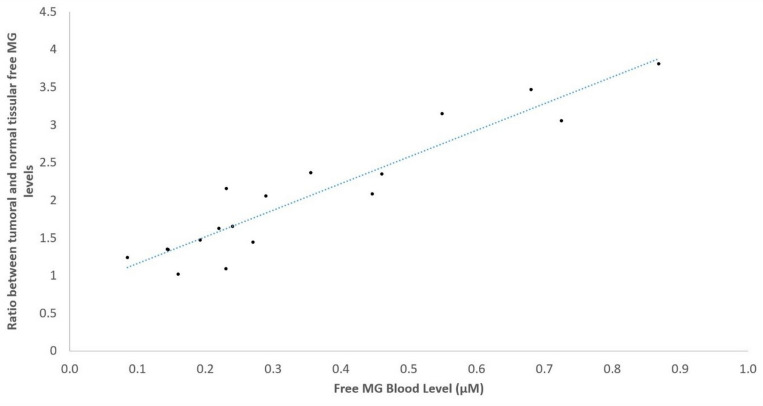
Positive correlation between free MG level tumor/normal tissue ratio and free MG level in the peripheral blood in 18 investigated cancer patients. The blue dotted line represent the linear regression line.

**Table 1 cancers-16-03922-t001:** Age and sex ratio in normal healthy controls or patients.

Demographic Data	Normal Healthy Controls	Normo-Glycemic Treated Type 2 Diabetes	Overall Cancers Patients	Hyperglycemic Type 2 Diabetes
*n*	68	12	139	10
Age (mean +/− SD)	45.32 +/− 15.61	62.33 +/− 10.79	62.01 +/− 10.76	55.50 +/− 15.61
Age (median [range])	43 [23–83]	60 [49–81]	62.5 [28–90]	67 [37–81]
Sex ratio (women/men)	41/27	7/5	68/71	3/7
Female (%)	60.3	58.3	48.92	30

**Table 2 cancers-16-03922-t002:** Mean free MG blood level values (μM), ± standard errors and confidence intervals in cancer patients; in comparison with (1) normal subjects and normo-glycemic treated type 2 diabetic patients, and (2) hyperglycemic untreated or relapsed type 2 diabetic patients (used as positive controls).

Type	*n*	Mean Free MG (µM)	*p* *	*p* **
Normal healthy Controls	68	0.0647 +/− 0.0031 [0.021–0.121]	_	_
Normo-glycemic treated type 2 Diabetes	12	0.0676 +/− 0.0281 [0.064–0.089]	0.046	_
Overall cancers patients	139	0.1758 +/− 0.0117 [0.035–0.868]	<0.0001	_
Hyperglycemic type 2 diabetes	10	0.1624 +/− 0.0074 [0.138–0.199]	<0.0001	0.965

*p*: probability that difference is due to random variation. * Comparison with the normally healthy control group for mean level values using the Wilcoxon rank sum test (α = 0.016). The Bonferroni correction was applied, which sets the α cut-off of significance at 0.05/3, i.e., 0.0016. ** Comparison between the hyperglycemic type 2 diabetes and the cancer groups of patients for mean level values using the Wilcoxon rank sum test (α = 0.05).

**Table 3 cancers-16-03922-t003:** Mean MG blood level values ± standard errors and confidence intervals (μM) in 139 cancer patients.

Cancer Type	*n*	Mean Free MG (µM)	*p* *
Bronchus	52	0.2354 +/− 0.0235 [0.085–0.868]	<0.0001
Glioblastoma	28	0.2201 +/− 0.0256 [0.068–0.814]	<0.0001
Other digestive tumors	8	0.1758 +/− 0.0371 [0.053–0.352]	0.0002
Colorectal	9	0.1666 +/− 0.0347 [0.038–0.352]	<0.0001
Pancreas	7	0.1496 +/− 0.0210 [0.099–0.248]	<0.0001
Gynecologic **	10	0.1479 +/− 0.0224 [0.053–0.223]	0.0004
Prostate	8	0.1127 +/− 0.0157 [0.081–0.217]	0.0003
Breast	12	0.0985 +/− 0.0114 [0.035–0.191]	0.002
Head and Neck	5	0.0980 +/− 0.0330 [0.040–0.210]	0.654
Total	139	0. 1758 +/− 0.0117 [0.035–0.868]	<0.0001

*p*: probability that difference is due to random variation. * Comparison between normally healthy controls and specific cancer groups of patients for mean level values using the Wilcoxon rank sum test (α = 0.005). The Bonferroni correction was applied, which sets the α cut-off of significance at 0.05/10; i.e., 0.005. ** Endometrial, ovarian and undetermined pelvic carcinomas.

**Table 4 cancers-16-03922-t004:** Free MG assessment mean values according to therapeutic response in 98 evaluable patients.

Therapeutic Response	*n*	Mean Free MG (µM) +/− Standard Error
Complete response	38	0.0610 +/− 0.0252
Partial response	31	0.1305 +/− 0.0721
Stable/progressive	29	0.1405 +/− 0.0617

## Data Availability

Due to privacy or ethical restrictions, study data can be made available to interested researchers upon request. Requests can be directed to data manager Philippe Irigaray (philippei.artac@gmail.com).

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
