# Peer review of "Free Methylglyoxal as a Metabolic New Biomarker of Tumor Cell Proliferation in Cancers"

_cancers, 2024, doi:10.3390/cancers16233922_

Round 1
Reviewer 1 Report
Comments and Suggestions for Authors
This manuscript proposes the use of methylglyoxal (MG) as new biomarker for cancer research, especially some types of cancers for which there is still no available biomarker. MG is a subproduct of the spontaneous chemical oxidation of glucose, and cancer cells often exhibit the Warburg effect and greater use of glucose. It is well known that MG is considered an AGE precursor, and its concentration is related to hyperglycemia and oxidative stress. This and other limitations are discussed correctly in the manuscript. Methods are also correct, and the difficulties stated at lines 123 and 365 can be easily solved.
MG is measured in biopsies of cancer and normal cells as well as peripheral blood. The last one could be more interesting for the follow-up of patients (total peripheral blood due to the difficulties for measuring plasma and intracellular erythrocyte.
Points to be addressed before definitive acceptance:
The number of samples measured seems to be 139 patients bearing various types of cancer at different TNM stages, and 68 healthy normal controls. However, free MG peripheral blood measurements was performed in 38 patients. Figure 1 states 130 samples. Figure 3 concerns only 107 cancer patients. Figure 4 is a positive correlation between free MG level tumor/normal tissue ratio and free MG level in the peripheral blood in only 18 evaluable cancer patients. The different number of samples could affect the statistical significance of some data. On the other hand, Figures 3 and 4 could be translated to supp. Material, as they are accessory to the main goal of the paper. The format of the figures could be improved to save room.
Conclusion at the abstract : Replace “Free MG measured in the blood is a new metabolic biomarker clinically useful …..” by “Free MG measured in the blood could be a new metabolic biomarker clinically useful… “. These results suggest this possibility, but further research should be made. Other statements, such as lines 176-177, are reasonable.
Concerning Supp. information
Lines 35-36 are unnecessary. This is not surprising due to the accumulation of glucose, oxygen and metal ion (Fe) in erythrocytes.
Define the abbreviation 2MQ
English would be revised for improving some expressions.
MG concentrations are nM at Figure S1, but mM at lines 30- 31 and Tables with data suggest micromolar range. Please, clarify.
Replace 25 by 30% (3 females in 10 hyperglycaemic patients) at the Table
|
Female (%) |
60.3 |
58.3 |
48.92 |
25 |
Lines 133-140 would be deleted. They seem to be requirements from the instructions to authors.
Author Response
Comment 1: The number of samples measured seems to be 139 patients bearing various types of cancer at different TNM stages, and 68 healthy normal controls. However, free MG peripheral blood measurements was performed in 38 patients. Figure 1 states 130 samples. Figure 3 concerns only 107 cancer patients. Figure 4 is a positive correlation between free MG level tumor/normal tissue ratio and free MG level in the peripheral blood in only 18 evaluable cancer patients. The different number of samples could affect the statistical significance of some data. On the other hand, Figures 3 and 4 could be translated to supp. Material, as they are accessory to the main goal of the paper. The format of the figures could be improved to save room.
Response 1: We’ve clarified page 4 that the different numbers of samples used for free MG measurement in case of TNM staging and therapeutic response, were due to a lack of clinical information. We also changed in figure 1, 130 samples by 133 (a mistake).
Finally, we suppressed Fig 3 which will be published elsewhere, but have kept Fig 4 in the text because we believed it is a very important biological original data which support our clinical data.
Comment 2: Conclusion at the abstract : Replace “Free MG measured in the blood is a new metabolic biomarker clinically useful …..” by “Free MG measured in the blood could be a new metabolic biomarker clinically useful… “. These results suggest this possibility, but further research should be made. Other statements, such as lines 176-177, are reasonable.
Response 2: We’ve replaced "is" by "could be" in the abstract.
Comment 3: Concerning Supp. information. Lines 35-36 are unnecessary. This is not surprising due to the accumulation of glucose, oxygen and metal ion (Fe) in erythrocytes.
Response 3: We decided to kept the sentence.
Comment 4: Concerning Supp. information. Define the abbreviation 2MQ
Response 4: In fact it is 2MQX, thank you for your observation. We’ve defined 2MQX in the supp method.
Comment 5: Concerning Supp. information. English would be revised for improving some expressions.
Response 5: Thanks. English was reviewed by a native.
Comment 6: Concerning Supp. information. MG concentrations are nM at Figure S1, but mM at lines 30- 31 and Tables with data suggest micromolar range. Please, clarify.
Response 6: Thank you again for your observation. It is a mistake in Figure S1. It is well µM instead of nM or mM in the text. Corrections were made.
Comment 7: Replace 25 by 30% (3 females in 10 hyperglycaemic patients) at the Table
|
Female (%) |
60.3 |
58.3 |
48.92 |
25 |
Response 7: We’ve also replaced 25% by 30% for the sex ratio of hyperglycemic type 2 diabetes. Thanks for your help.
Comment 8: Lines 133-140 would be deleted. They seem to be requirements from the instructions to authors.
Response 8: The lines were deleted.
Reviewer 2 Report
Comments and Suggestions for Authors
In cancers-3293593, Belpomme et al report free methylglyoxal as a new metabolic biomarker of tumor cell proliferation. The topic of the manuscript is interesting and fits well the scope of the journal. The reviewer feels it can be accepted after some minor amendments.
(1) Table 1: The age of normal healthy controls is much younger than other groups. Is methylglyoxal correlated to age?
(2) The bioanalytical method or methylglyoxal should be validated properly. Any supporting evidence?
(3) The authors used t-test to analyze their data. Have they checked whether the distribution is normal?
(4) Any difference between female and male?
Author Response
In cancers-3293593, Belpomme et al report free methylglyoxal as a new metabolic biomarker of tumor cell proliferation. The topic of the manuscript is interesting and fits well the scope of the journal. The reviewer feels it can be accepted after some minor amendments.
Comment 1: Table 1: The age of normal healthy controls is much younger than other groups. Is methylglyoxal correlated to age?
Response 1: MG is not correlated to age. This was indicated in the text page 3 line 95 in Materials and Methods. Nevertheless, we added the ρ and p values.
Comment 2: The bioanalytical method or methylglyoxal should be validated properly. Any supporting evidence?
Response 2: The bioanalytical method of MG has been well validated: this was indicated in line 133 in Materials and Methods. This method is the one proposed by Rabbani and Thornalley, reference 16 of the paper.
Comment 3: The authors used t-test to analyze their data. Have they checked whether the distribution is normal?
Response 3: We agree with you. After verification the distribution was not normal. So we replaced the student t-test by the Wilcoxon rank sum test. This explain why several p-values were different, but others were not.
Comment 4 : Any difference between female and male?
Response 4: We did not check for differences between female and male, because many cancer localisations were already different according to the sex. For those occurring in both sexes (bronchus and digestive tumors) there were no differences. It’s indicated in the text page 3 line 97 in Materials and Methods..
Reviewer 3 Report
Comments and Suggestions for Authors
The manuscript by Belpomme et al. reports that blood levels of methylglyoxal may serve as a new biomarker for tumour cell proliferation in cancer. In their study, the authors found increased levels of methylglyoxal in the peripheral blood of patients with different types of cancer, while positive response to cancer treatment was associated with decreased levels of methylglyoxal in the peripheral blood. Thus, the results of this study suggest a direct practical application for the major metabolite of the Warburg effect. The manuscript is well written and easy to follow. However, I would like to suggest that the authors include the information on the methylglyoxal levels in the peripheral blood of the patients who responded positively to the treatment in the abstract and discuss this issue more in the discussion from the point of view of a new tool that offers a new possibility to monitor the efficacy of therapy and possible tumour remission.
Author Response
Comment: I would like to suggest that the authors include the information on the methylglyoxal levels in the peripheral blood of the patients who responded positively to the treatment in the abstract and discuss this issue more in the discussion from the point of view of a new tool that offers a new possibility to monitor the efficacy of therapy and possible tumour remission.
Response: We agree with you. As recommended, we included in the abstract and the discussion that MG measurement could be a new tool to monitor cancer treatment.